# Body Mass Index, Physical Activity, Cardiorespiratory Endurance and Quality of Life among Children with Physical Disabilities

**DOI:** 10.3390/children10091465

**Published:** 2023-08-28

**Authors:** Nimale Supramaniam, Asfarina Zanudin, Nor Azura Azmi

**Affiliations:** 1Physiotherapy Program, Centre for Rehabilitation and Special Needs Studies, Faculty of Health Science, Universiti Kebangsaan Malaysia, Jalan Raja Muda Abdul Aziz, Kuala Lumpur 50300, Selangor, Malaysia; p97070@siswa.ukm.edu.my (N.S.); asfarina.zanudin@ukm.edu.my (A.Z.); 2Physiotherapy Unit, Hospital Tuanku Ampuan Najihah Kuala Pilah, Kuala Pilah 72000, Negeri Sembilan, Malaysia

**Keywords:** physical disabilities, functional activity, physical activity, cardiorespiratory endurance, obesity, body mass index

## Abstract

Background: Children with physical disabilities (PD) have reduced levels of physical activity (PA) compared to typically developing children, which increases their risk of becoming overweight and obese, which leads to numerous adverse health consequences. This study aimed to determine the differences between groups classified by body mass index (BMI) percentile in terms of PA levels, cardiorespiratory endurance and quality of life (QoL), and also to evaluate the relationship between BMI percentile and PA levels, cardiorespiratory endurance and QoL in children and adolescents with physical disabilities. Methods: A total of 172 children and adolescents with PD aged between 5 and 17 years from Hospital Tunku Azizah were included in this cross-sectional study. The BMI percentile was calculated to determine the weight status. PA levels were assessed with the Physical Activity Questionnaire for Older Children (PAQ-C), cardiorespiratory endurance was measured by the Six-Minute Walk Test (6MWT) and QoL was measured by the Paediatric Quality of Life Inventory version 2.0 (PedsQL 2.0). Results: According to the BMI percentile, 70.3% had a healthy BMI percentile (50th to 84th percentile), 11.6% were overweight (50th to 84th percentile), 11% were underweight (0–49th percentile) and 7.0% were obese (95th percentile and above). Most children reported a healthy weight, and the rates of being overweight and obese were higher in children who could ambulate without aids (6.4% and 3.5%, respectively) compared to those who used walking aids (5.2% and 3.5%, respectively). Significant differences were found in the PAQ-C, 6MWT and PedsQL 2.0 scores between different BMI percentile groups (*p* < 0.05). There were also significant correlations between the BMI percentile and the PAQ-C (*r* = 0.209, *p* < 0.001), 6MWT (*r* = 0.217, *p* < 0.001) and PedsQL 2.0 (*r* = 0.189, *p* < 0.001). Conclusion: The rate of being overweight and obese is greater among children who ambulate without aids than among those with aids. An increase in the BMI percentile can reduce the QoL in different ways. This study suggests that children with PD who can walk without aids are at a greater risk of being overweight and obese. Hence, the engagement of this population in PA is crucial for their weight management.

## 1. Introduction

Childhood obesity has reached widespread prevalence in the past 30 years [1], and rates of childhood obesity have reportedly increased over the years. In Malaysia, up to 30% of children were found to be overweight or obese [2]. Obesity in typically developing peers leads to numerous health and social repercussions such as self-isolation, low self-esteem and reduced quality of life (QoL) that frequently persist into adulthood [3]. Obesity is not only a risk factor for chronic diseases like hypertension, diabetes and hyperlipidaemia, but also for developing secondary complications such as muscle deconditioning, low endurance and mobility limitation, and it is associated with depression and social isolation [4]. This is no exception in children with physical disabilities (PD) [5]. 

Despite the fact that children with PD are known to have reduced oral intake, inadequate nutrition and physical and mental growth retardation due to cognitive impairment [6], the overweight and obesity prevalence among them was significantly higher, which was three and six times higher than in typically developing peers (*p* < 0.001), as reported in a study of Dutch children [7]. A cohort study has reported that 19.4% of 587 ambulatory PD children were overweight and obese [8]. Moreover, several studies have reported that children with milder forms of disability and higher levels of functioning have higher rates of obesity than those with moderate or severe forms of disabilities [9]. In addition, children with mild disabilities have lower rates of motor control and neuromuscular disorders that impact their daily activities than children with moderate and severe disabilities [10]. However, children with PD have a lower energy expenditure and limited participation in age-appropriate physical and athletic activities compared to their normal peers [11]. Consequently, these children, who are less mobile than healthy children, are at a higher risk of obesity. Being overweight and obese further complicates the performance of daily activities and can negatively impact overall health by causing decreased physical activity (PA) and inadequate self-care [12].

In the paediatric population, PA plays a key role in preventing excessive body mass. Children who engage more in PA generally have less body fat than those who are less engaged in PA; however, a significant proportion of this population do not meet the PA recommendations [13]. The World Health Organization (WHO) Global Physical Activity Guidelines recommend that children engage in at least 60 min of moderate-intensity physical activity (MVPA) per day. These guidelines also apply to children with disabilities [14]. However, in 2021, the Department of Health and Social Care, United Kingdom, reviewed the evidence on PA pertaining to disabled children and adolescents aged 5 to 17 years old, and published the guidelines to improve their physical and mental health (15). The new guidelines by the UK government recommended that these children should engage in at least 120 to 180 min of MVPA per week, which includes various forms of PA, such as walking and cycling, as well as exercising 20 min per day or 40 min three times per week [15]. The characteristics of prominent PD in children include abnormal muscle tone, poor muscle control, weakness and retarded physical development, which are risk factors for physical inactivity and lead to obesity. Physical inactivity is a significant contributor to the increased likelihood of obesity in children [16,17]. Therefore, it is important to encourage children to reduce sedentary lifestyles and engage more in PA. A focused and appropriate intervention is crucial for promoting PA and encouraging an active lifestyle to build a healthy population [18].

Moreover, higher levels of cardiorespiratory endurance during childhood are associated with an ideal body mass index (BMI) and lower cardiometabolic risk in children and adults [19]. It is important to note that cardiorespiratory endurance in childhood is a significant predictor of cardiovascular health in adolescence and adult life [20]. Given that low PA levels dictate cardiorespiratory fitness and muscle strength in overweight and obese individuals, this is particularly concerning. An increase in the PA level can benefit respiratory health and improve endurance in the long term [20].

The adverse health consequences of being overweight and obese extend beyond physical health. Overweight and obese children experience problems such as low self-esteem and confidence, depression, stigmatisation and reduced social interaction, which can negatively impact their psychological and social health [21]. An insufficient amount and energy consumption is required to enable and maintain mobility in children with PD, as well as muscle weakness, abnormal muscle tone and nutritional problems resulting from various factors that can alter health status [22]. Impaired physical and social functioning also has a detrimental impact on daily activities and leads to a reduction in the QoL. This is common in underweight, overweight or obese children. A diverse range of PA in a compelling atmosphere improves children’s socio-emotional development [23]. 

There has been scarce evidence reporting the influence of the BMI percentile on PA levels, cardiorespiratory endurance and QoL among children with PD. Identifying this influence is crucial to enlighten the government, which can be used for the development of public policies that strategize to increase PA and improve health, specifically in children with PD. Hence, this study aimed to determine the differences in PA levels, cardiorespiratory endurance and QoL among groups classified by BMI percentile in children with PD. We also aimed to explore the relationships between BMI percentile and PA level, cardiorespiratory endurance and QoL. We hypothesise that significant differences exist in these measures between groups classified by BMI percentile, and that there are relationships between BMI percentile and PA level, cardiorespiratory endurance and QoL.

## 2. Materials and Methods

### 2.1. Participants

This was a cross-sectional study of children with PD aged 8 to 17 years from Hospital Tunku Azizah, Kuala Lumpur. The convenience sampling method was used to recruit the participants in this study. Participants included in this study were children diagnosed with one of the following conditions: cerebral palsy (CP), spinal cord injury, Down syndrome, traumatic brain injury and stroke, who were able to ambulate with or without walking aids and were able to follow a simple command. Children with underlying pathologies, such as a tumour, inability to understand English or Malay languages, attention deficit hyperactivity disorder (ADHD), mental illness and autistic conditions were excluded from this study. The sample size was calculated using the Krejcie and Morgan (1970) formula. The population size for all children with PD in Hospital Tunku Azizah, Kuala Lumpur was 260. The sample size was calculated with a 95% confidence interval, 5% margin of error, and a 10% dropout rate. The calculated sample size was *n* = 172. 

### 2.2. Instruments

The weight and height of participants were measured using a standardised weighing–height scale (Jiangsu Suhong Medical Instruments Co., Ltd., Changzhou, China, model RGX-160), which was calibrated before each use against a standard weight. The BMI percentile was used to determine weight status. The BMI, calculated by dividing the body weight by the square of the height value, was determined using the corresponding value on the percentile curves developed separately by gender. In the study context, the children were categorised into four groups: underweight (0–49th percentile), healthy weight (50th to 84th percentile), overweight (85th to 94th percentile) and obese (95th percentile and above) [24]. 

The Six-Minute Walking Test (6MWT) is a standardised walking test used to evaluate the cardiorespiratory endurance in children with PD in this study. This test measures the participant’s walking distance within six minutes with the unit in metres [25]. The 6MWT is a valid and reliable tool for measuring the functional ability and cardiorespiratory endurance in children with PD. This test is easy to apply in the clinical setting and has been recommended as a submaximal test for children with CP at Gross Motor Function Classifications System (GMFCS) levels I to III. The 6MWT demonstrated excellent test–retest reliability among children with CP classified in GMFCS levels I to III aged 4–18 years old (ICC = 0.98) and 11–17 years old (ICC = 0.98) by [26,27,28]. During the 6MWT, the Borg scale was used to measure the overall level of dyspnoea and fatigue; it begins at zero, where breathing causes no difficulty, and progresses to 10, where breathing difficulty is at its maximum. 

The Physical Activity Questionnaire for Older Children (PAQ-C) was used to evaluate PA levels. It is a self-administered, seven-day recall questionnaire that contains nine items for PA level measurement and is valid to be used in children between the ages of 8 and 14 years old. However, one question that asked participants if they were feeling ill or if they had anything that prevented them from completing regular PA was not included in the overall score calculation. This questionnaire was translated into a Malay version and validated on 73 children, which showed that this version had good internal consistency with a Cronbach’s alpha value of 0.75–0.77, and the Spearmen’s correlation coefficient with 3-day physical activity recall (3DPAR) (*r* = 0.60, *p* < 0.01) was good among older children aged 10–17 years [29]. Consistent with previous studies [30,31], Cronbach’s alpha of total PA was satisfactory, ranging between 0.737 and 0.930, indicating that this questionnaire is suitable for evaluating PA in this population. The PAQ-C mean score was categorised as “low” (1–2.33), “moderate” (2.34–3.66) or “high” (3.67–5.00) [28].

The parent-reported Paediatric Quality of Life Inventory version 2.0 (PedsQL 2.0) is a modular instrument used to measure health-related QoL in children and adolescents aged 2–18 years. This questionnaire is a multidimensional parent proxy-reported scale and consists of 36 items with eight sub-dimensions. The first six domains measure parent self-functioning (i.e., physical functioning, emotional functioning, cognitive functioning, social functioning communication and worry), while the last two sub-dimensions measure family functioning (i.e., daily activities and family relationships), and are applicable to all paediatric populations [32]. The cumulative PedsQL 2.0 score was calculated by dividing the sum of all 36 items by the number of questions answered. The parent HRQOL summary scores were measured as the sum of items in the physical, emotional, social and cognitive functioning scales, divided by the number of items answered. The Family Functioning Scale was calculated as the sum of the items for daily activities and family relationships, divided by the number of items answered. The PedsQL 2.0 has been translated to the Malay version and validated [33]. Good reliability and validity were reported in a study among 383 caregivers of children with learning disabilities [33] (Cronbach’s alpha > 0.7), goodness-of-fit (2(426) = 878.842), *p* < 0.001; RMSEA = 0.053; CFI = 0.918; 2/df = 2.063. Each item was scored on a 5-point Likert scale ranging from 0 (never) to 4 (almost always), and the scale was reverse scored and linearly transformed to a 0–100 scale (0 = 100, 1 = 75, 2 = 50, 3 = 25 and 4 = 0); the higher the score, the greater the participants’ QoL [32].

### 2.3. Study Procedures

Data collection was conducted in the Physiotherapy Unit, Hospital Tunku Azizah, Kuala Lumpur, from March 2021 to May 2022. Informed consent was obtained from the parents and children prior to data collection. The socio-demographic information of the study participants, including age, gender, height, weight as well as the type and severity of PD, were collected. All children completed a self-administered PAQ-C questionnaire prior to their cardiorespiratory endurance assessment. The children were then instructed to complete the 6MWT, which was repeated three times with a five-minute break in between. Finally, their parents were required to fill up the PedsQL 2.0 questionnaire in order to complete the study assessment. All data were recorded in the assessment form. 

### 2.4. Statistical Analysis

Data were analysed using the Statistical Package for Social Science (SPSS) software version 23. The socio-demographic data were analysed using descriptive statistics. Data were presented as frequency, percentage and mean ± SD. Prior to data analysis, a normality test was conducted to examine the distribution of the data. A non-parametric test was chosen due to data not being normally distributed. The Kruskal–Wallis test was used to determine the differences in PA levels, cardiorespiratory endurance and QoL. The Spearman rho analysis was used the determine the correlation between BMI percentile and PA levels, cardiorespiratory endurance and QoL. The Kruskal–Wallis post-hoc test was used to determine significant differences among the groups. The statistical significance level was set at *p* < 0.05. 

## 3. Results

### 3.1. Sociodemographic Data and Evaluation of the BMI Percentile

Table 1 shows the socio-demographic characteristics of the participants. A total of 172 children with PD with a mean age of 10.82 ± 2.27 years, and 96 (55.8%) females and 76 (44.2%) males, participated in this study. The majority of the participants were children with Cerebral Palsy (CP), 98 (57%), followed by traumatic brain injury (TBI), 32 (18.6%), Down syndrome (DS), 23 (13.4%), stroke, 17 (9.9%) and spina bifida, 2 (1.2%). More than half of the participants, 100 (58.1%), use walking aids for ambulation, while 72 (41.9%) children do not. Classifications according to the BMI percentile revealed that 121 (70.3%) children were of normal weight, 20 (11.6%) were overweight, 19 (11%) were underweight and 12 (7%) were obese. The prevalence of healthy weight was found to be higher among children with PD who ambulated using walking aids (69%), compared to children who ambulated without walking aids (6.4% and 3.5%, respectively) (Table 2) in this study. 

### 3.2. Differences in the PAQ-C Score, 6MWT and the Total and All Sub-Dimensions of the PedsQL 2.0 between Groups by BMI Percentile

The Kruskal–Wallis test revealed significant differences in the total PAQ-C score, 6MWT and all of the sub-dimensions of the PedsQL 2.0 between children with PD who were underweight, healthy weight, overweight and obese (*p* < 0.05) (Table 3). Significant differences were also found between children who were underweight and a healthy weight in terms of the total PAQ-C, 6MWT, total PedsQL 2.0 and all the sub-dimensions of the PedsQL 2.0. Underweight and overweight children reported statistical differences in terms of the PAQ-C and two sub-dimensions of the PedsQL 2.0 (i.e., worry and daily activities), and obese and healthy weight children showed differences in one of the PedsQL 2.0 domains (i.e., physical functioning) (*p* < 0.05). 

### 3.3. Correlation between the BMI Percentile and the PAQ-C, 6MWT and PedsQL

The Spearmen correlation test showed a positive correlation between the BMI percentile and the PAQ-C and 6MWT (*p* < 0.05) (Table 4). Positive correlations were also found between the BMI percentile and the total score of the PedsQL 2.0, as well as the PedsQL 2.0 sub-dimensions, such as social functioning, communication, worry, daily activities, family relationships, parent HRQOL functioning and family functioning (*p* < 0.05). There was no correlation observed between the BMI percentile and the three sub-dimensions of the PedsQL 2.0, which are physical functioning, emotional functioning and cognitive functioning (*p* > 0.05) (Table 5). 

## 4. Discussion

The main findings in this study were as follows: the rate of being overweight and obese was higher in children who ambulated without walking aids than those who ambulated using walking aids; there was a difference between the groups classified by the BMI percentile children in terms of the PA level, cardiorespiratory endurance and QoL; and there was a weak significant correlation between the BMI percentile and PA level, cardiorespiratory endurance and QoL. These findings imply that the policies introduced by the government to prevent overweight and obesity in children with PD need to be tailored to their physical abilities, such as providing more gymnasiums that are friendly for children with special needs and providing easy access for parents and children throughout the country. 

Although studies have reported that underweight rates are greater in children with PD due to insufficient nutrition, studies have also shown that obesity and overweight rates are higher in children with PD, particularly ambulatory children, as well as in typically developing children [9]. Similar to these studies, in this study, the rate of overweight and obesity were higher in children who ambulated without walking aids compared to those who ambulated using walking aids. Moreover, it was discovered that the risk of obesity was greater among ambulatory children with low severity disabilities, whereas the rate of being underweight was higher among children with high severity disabilities and non-ambulatory [34]. The high risk of overweight and obesity in ambulatory children with PD is driven by high fat accumulation relative to musculoskeletal deficits [35].

In this study, there were differences in the PA level (PAQ-C), cardiorespiratory endurance (6MWT) and QoL (PedsQL 2.0) between the groups classified by BMI percentile. Studies have been conducted to examine the PA levels, cardiorespiratory endurance and QoL among children with PD, and it has been reported that the majority of them who regularly participated in PA reduced their risk of being overweight and obese, which consequently improved their cardiorespiratory endurance and QoL [36,37,38]. Children who were underweight had the lowest scores on the PAQ-C, 6MWT, total PedsQL 2.0 and all sub-dimensions of the PedsQL 2.0. Similar findings were observed among children who were overweight and obese. Meanwhile, children with a healthy weight had the highest score on the PAQ-C, 6MWT, total PedsQL 2.0 and all the sub-dimensions of the PedsQL 2.0. Increasing PA among children with PD reduces the BMI and body mass fat while increasing muscle mass [36]. Overweight and obese children have lower levels of PA compared to their non-overweight and non-obese peers. Thus, they have fewer opportunities to practice their motor skills, limiting their capacity to participate and causing muscle deconditioning. A previous study conducted on Taiwanese children found that healthy, underweight and overweight children had significantly higher cardiorespiratory endurance levels than obese children [39]. A systematic review has reported that children who are overweight or obese have a lower QoL than children who are of a healthy weight, and that QoL deteriorates as body mass index increases [40]. Nevertheless, psychosocial-related quality of life was not significantly associated with obesity in children, except for the social functioning subscale, which showed a strong association [41].

This study demonstrated a weak positive correlation between BMI percentile and PA level, indicating that as PA participation increases, so does the risk of being overweight and obese. However, a previous study indicated that PA was negatively associated with the risk of obesity [42]. Studies have shown that the risk of being overweight and obese in children who met the minimum requirement of 60 min of MVPA per day increased by 7% as the number of days in which children were physically active increased. Therefore, if overweight and obese children engage in MVPA as recommended, their risk of being overweight and obese is reduced by 49% [42]. Regular participation in PA reduces obesity [43,44]. Conversely, a study on the relationship between BMI and PA conducted on children in public schools in Granada found that students who reported being physically active for more than three hours a week were more likely to be overweight or underweight than those who did not meet the above criteria [45]. However, no significant correlation was found between obesity and PA in a study of Down syndrome children [46]. Developing intervention programs to promote physical activity, beginning in childhood, is important to prevent overweight and obesity, resulting in a decreased risk of NCDs among children later in life.

In this study, a weak positive correlation between BMI percentile and cardiorespiratory endurance was identified. Several studies have reported that high cardiorespiratory endurance may reduce the health risks associated with being overweight and obese in children [20,47]. However, being overweight and obese has also been reported to be inversely correlated (*p* < 0.001) with cardiorespiratory endurance [37,48,49]. Children and adolescents with low cardiorespiratory endurance had a greater percentage of body fat, a higher risk of high blood pressure and metabolic syndrome, and consequently, a lower QoL [50]. A better understanding of the relationship between cardiorespiratory endurance and the BMI percentile is essential for the development of intervention programs designed to prevent obesity.

The BMI percentile was positively correlated with all the sub-dimensions of the PedsQL 2.0 including communication, worry, daily activities, family relationship, parents HRQoL, family functioning summary and the total PedsQL 2.0 score in this study. In parallel to this study, children who participated in organised sport activity more than three hours daily reported better intrapersonal and interpersonal levels, emotional intelligence and stress management than those who did not [51]. A very different result has been found by Melguizo-Ibáñez et al., 2022, who reported a negative association between PA and emotional intelligence among female elementary school students [52]. A previous study among obese children found a negative association between the degree of obesity and quality of life [53]. In line with the findings of a previous study, it was discovered that the proportion of body fat had a stronger association with psychosocial functioning than BMI [54]. A comprehensive study in a paediatric population reported that BMI has a moderate-to-strong negative association with overall QoL. This review indicated that the total QoL score was negatively associated with BMI, waist circumference and weight [55]. The BMI percentiles of children with PD are significant indicators of health status and a crucial factor that has a substantial impact on wellbeing, and consequently, QoL.

To our knowledge, this is the first study conducted in Malaysia, and the findings will be used for future studies in the field of PD, such as the development of the BMI percentile growth chart suitable for children with PD. The majority of participants in this study were a healthy weight based on their BMI percentile. However, BMI percentile analysis suggests that children with PD have a high rate of being overweight and obese, particularly among children without walking aids. The results of this study confirmed that the BMI percentile plays an essential role in PA involvement, cardiorespiratory endurance and QoL among children with PD. The BMI percentile can have an impact on the daily activities and QoL of children with PD of varying severities and disabilities. The findings in this study will encourage the development, strengthening and implementation of strategies and action plans that are sustainable, comprehensive and actively involve all sectors, including the government and the private sector. Early height and weight measurements may help these children obtain sufficient energy to sustain mobility and QoL in adolescence and adulthood. Precautions and regular monitoring should be taken from an early stage. It is recommended that further research be conducted in this field.

This study has a few limitations that should be considered. This study had a small sample size, and the participants were recruited from a single cohort of children with PD from one hospital. Therefore, the findings may not be generalisable to all Malaysian children with PD. In order to obtain more representative and generalisable results, future studies should recruit a large sample from various hospitals and rehabilitation centres in Malaysia. Besides that, the WHO BMI percentile chart was developed for children with normal growth rates and not for children with PD.

## 5. Conclusions

In conclusion, this study indicates that children who ambulated without aids had a higher risk of being overweight and obese than in those with aids. Being overweight and obese can reduce the level of daily activity, cardiorespiratory endurance and QoL in various ways. Height and weight measurements during childhood may be used to determine whether children with PD will have sufficient energy to continue walking and achieve functional independence as they grow into adolescence and adulthood. Early prevention measures should be implemented at the individual, family, school and community level to reduce the risk of overweight and obesity. It can be implemented with existing intervention strategies and approaches, and by making appropriate modifications.

## Figures and Tables

**Table 1 children-10-01465-t001:** Demographic characteristics of the participants (*n* = 172).

Age (years), mean ± SD	10.82 ± 2.27
Gender, *n* (%)	
Male	76 (44.2)
Female	96 (55.8)
Usage of Walking Aids, *n* (%)	
Without Walking Aids	72 (41.9)
With Walking Aids	100 (58.1)
Diagnosis, *n* (%)	
Cerebral Palsy	98 (56.9)
DS	23 (13.4)
TBI	32 (18.6)
Spina Bifida	2 (1.2)
Stroke	17 (9.9)

DS: Down syndrome; TBI: traumatic brain injury; mean ± SD: mean ± standard deviation.

**Table 2 children-10-01465-t002:** BMI percentile values according to usage of walking aids.

BMI Percentile	Without Walking Aids (*n* = 72) *n* (%)	With Walking Aids (*n* = 100) *n* (%)	Total (*n* = 172) *n* (%)
<5th underweight	3 (1.7)	16 (9.3)	19 (11.0)
5th–85th healthy weight	52 (30.2)	69 (40.2)	121 (70.4)
85th–94th overweight	11 (6.4)	9 (5.2)	20 (11.6)
≥95th obese	6 (3.5)	6 (3.5)	12 (7.0)

BMI: body mass index.

**Table 3 children-10-01465-t003:** Difference in the PAQ-C score, 6MWT and the total and all sub-dimensions of the PedsQL 2.0 between groups by BMI percentile (*n* = 172).

	Underweight (*n* = 19)	Healthy Weight (*n* = 121)	Overweight (*n* = 20)	Obese (*n* = 12)		
	Mean ± SD	Mean ± SD	Mean ± SD	Mean ± SD	χ^2^	*p*
PAQ-C score	1.91 ± 0.11	2.62 ± 0.07	2.49 ± 0.10	2.17 ± 0.16	18.32	0.000 *
6MWT	231.44 ± 5.95	266.01 ± 4.10	257.92 ± 5.31	246.92 ± 7.34	12.81	0.005 *
Total PedsQL	33.59 ± 2.05	47.10 ± 1.29	43.06 ± 1.81	246.92 ± 7.34	22.05	0.000 *
Physical functioning	42.11 ± 2.81	57.68 ± 1.43	50.63 ± 0.92	45.83 ± 2.81	23.31	0.000 *
Emotional functioning	39.74 ± 2.85	53.68 ± 1.38	47.50 ± 1.72	45.83 ± 2.81	20.69	0.000 *
Social functioning	37.50 ± 2.70	51.34 ± 1.33	45.00 ± 1.90	43.75 ± 2.88	17.36	0.001 *
Cognitive functioning	35.53 ± 2.91	48.22 ± 1.43	42.50 ± 2.63	41.67 ± 3.55	13.04	0.005 *
Communication	32.89 ± 2.51	44.22 ± 1.30	42.50 ± 2.63	40.28 ± 3.53	11.68	0.009 *
Worry	27.11 ± 1.89	41.03 ± 1.46	41.25 ± 2.74	31.67 ± 2.97	20.99	0.000 *
Daily activities	25.00 ± 1.91	39.53 ± 1.54	37.50 ± 2.87	29.17 ± 2.81	19.79	0.000 *
Family relationship	24.21 ± 2.07	35.67 ± 1.39	34.00 ± 2.60	27.08 ± 2.78	15.08	0.002 *
Parent HRQOL Summary	38.72 ± 2.62	52.73 ± 1.30	46.41 ± 1.60	44.27 ± 2.80	20.47	0.000 *
Family Functioning Summary	24.61 ± 1.95	37.60 ± 1.42	35.75 ± 2.59	28.13 ± 2.70	19.89	0.000 *

* Kruskal–Wallis test, χ^2^ test, *p* < 0.05; mean ± SD; mean ± standard deviation; PAQ-C: Physical Activity Questionnaire for Older Children; 6MWT: 6-Min Walking Test; PedsQL 2.0: Paediatric Quality of Life Inventory version 2.0; HRQOL: health-related quality of life; BMI: body mass index.

**Table 4 children-10-01465-t004:** Correlations between the BMI percentile and the PAQ-C and 6MWT (*n* = 172).

	BMI Percentile		PAQ-C Score
PAQ-C score	r	0.209 *	
	*p*	0.006	
6MWT	r	0.217 *	0.927 *
	*p*	0.004	0.000

* *p* < 0.05, Spearman’s rho; BMI: body mass index; PAQ-C: Physical Activity Questionnaire for Older Children; 6MWT: 6-Min Walking Test.

**Table 5 children-10-01465-t005:** Correlations between the BMI percentile and the total all sub-dimensions of the PedsQL 2.0 (*n* = 172).

	BMI Percentile		Total PedsQL	PF	EF	SF	CF	Com	Wry	DA	FR	Parents HRQOL	FF
Total PedsQL	r	0.189 *											
	*p*	0.013											
PF	r	0.147	0.918										
	*p*	0.054	0.000										
EF	r	0.143	0.841 *	0.884									
	*p*	0.062	0.000	0.000									
SF	r	0.152 *	0.910 *	0.813 *	0.873								
	*p*	0.047	0.000	0.000	0.000								
CF	r	0.138	0.888 *	0.742 *	0.765 *	0.961 *							
	*p*	0.070	0.000	0.000	0.000	0.000							
Com	r	0.196 *	0.926 *	0.753 *	0.734 *	0.902 *	0.938 *						
	*p*	0.010	0.000	0.000	0.000	0.000	0.000						
Wry	r	0.201 *	0.913 *	0.752 *	0.659 *	0.760 *	0.766 *	0.874 *					
	*p*	0.008	0.000	0.000	0.000	0.000	0.000	0.000					
DA	r	0.180 *	0.902 *	0.782 *	0.664 *	0.724 *	0.717 *	0.802 *	0.931 *				
	*p*	0.018	0.000	0.000	0.000	0.000	0.000	0.000	0.000				
FR	r	0.200 *	0.900 *	0.894 *	0.714 *	0.714 *	0.681 *	0.734 *	0.811 *	0.870 *			
	*p*	0.009	0.000	0.000	0.000	0.000	0.000	0.000	0.000	0.000			
Parent HRQOL	r	0.157 *	0.975 *	0.941 *	0.862 *	0.933 *	0.911 *	0.898 *	0.824 *	0.818 *	0.869 *		
	*p*	0.040	0.000	0.000	0.000	0.000	0.000	0.000	0.000	0.000	0.000		
FF	r	0.186 *	0.936 *	0.878 *	0.722 *	0.715 *	0.730 *	0.799 *	0.900 *	0.963 *	0.962 *	0.881 *	
	*p*	0.014	0.000	0.000	0.000	0.000	0.000	0.000	0.000	0.000	0.000	0.000	

* *p* < 0.05, Spearman’s rho; BMI: body mass index; PedsQL 2.0: Paediatric Quality of Life Inventory version 2.0; PF: physical functioning; EF: emotional functioning; SF: social functioning; CF: cognitive functioning; Com: communication; Wry: worry; DA: daily activities; FR: family relationship; HRQOL: health-related quality of life; FF: family functioning.

## Data Availability

The datasets used and/or analysed during the current study are available from the corresponding author on reasonable request.

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
