# Peer review of "Body Mass Index, Physical Activity, Cardiorespiratory Endurance and Quality of Life among Children with Physical Disabilities"

_children, 2023, doi:10.3390/children10091465_

Round 1

Author Response

Dear Reviewer,

Thank you very much for your comments  and please see the attachment for the response. Thank you.

Reviewer 2 Report

First of all I would like to congratulate the authors of this research, however it is necessary to improve the research so that it can be published. 

First of all, adapt the research to the journal's template. Likewise, the bibliography should be adapted to the journal's standards. 

The theoretical framework is well contextualized, however I suggest adding the following research: 

Melguizo-Ibáñez, E.; González-Valero, G.; Badicu, G.; Filipa-Silva, A.; Clemente, F.M.; Sarmento, H.; Zurita-Ortega, F.; Ubago-Jiménez, J.L. Mediterranean Diet Adherence, Body Mass Index and Emotional Intelligence in Primary Education Students-An Explanatory Model as a Function of Weekly Physical Activity. Children 2022, 9, 872. https://doi.org/10.3390/children9060872. https://doi.org/10.3390/children9060872

Melguizo-Ibáñez, E.; González-Valero, G.; Puertas-Molero, P.; Alonso-Vargas, J.M. Emotional Intelligence, Physical Activity Practice and Mediterranean Diet Adherence-An Explanatory Model in Elementary Education School Students. Children 2022, 9, 1770. https://doi.org/10.3390/children9111770. https://doi.org/10.3390/children9111770

Amado-Alonso, D.; León-del-Barco, B.; Mendo-Lázaro, S.; Sánchez-Miguel, P.A.; Iglesias Gallego, D. Emotional Intelligence and the Practice of Organized Physical-Sport Activity in Children. Sustainability 2019, 11, 1615. https://doi.org/10.3390/su11061615

Jolić Marjanović, Z.; Altaras Dimitrijević, A.; Protić, S.; Mestre, J.M. The Role of Strategic Emotional Intelligence in Predicting Adolescents’ Academic Achievement: Possible Interplays with Verbal Intelligence and Personality. Int. J. Environ. Res. Public Health 2021, 18, 13166. https://doi.org/10.3390/ijerph182413166

Likewise, the research presents a reliable analysis of the data and a consistent and well-developed methodological framework. The instruments used have been previously validated by the scientific community. 

Finally, I suggest revising the wording of the research and revising the English.

English redaction must be improved. 

Author Response

(The authors gave the same response as above.)

Reviewer 3 Report

Dear Authors,

Thank you for the opportunity to review this manuscript. In my opinion, the publication requires minor corrections and additions.

1. Keywords should be different than in the title - words in the title are already keywords.

2. Please describe in detail what standards and ranges of standards you have taken into account? I find it very debatable that you set standards based on your material for both girls and boys combined - is that it? For children, BMI norms are more sensitive than for adults. Please respond to the methodological problem.

3. Test values should be provided in the tables, e.g. tab. 3 - no Kruskal-Wallis test value.

4. In the tables, the % values do not add up to 100 - eg in table 2 - please correct it.

5. In my opinion, all analyzes must be done separately for boys and girls - just as the norms are separate for the sex of children.

6. Editorial errors, spelling errors - to be corrected.

Kind regards,

revewier

Author Response

(The authors gave the same response as above.)

Round 2

Reviewer 1 Report

Dear Authors

Congratulations on the adjustments presented. The changes made significantly contributed to the improvement of the manuscript. I must point out that the quality of the text presented is satisfactory. I forwarded my opinion to the Editor.

Best Regards,

The Reviewer.

Reviewer 2 Report

The authors have included all proposed improvements. It can be published